# Spin Hall and Edelstein effects in chiral non-collinear altermagnets

Mengli Hu [1], Oleg Janson [1], Claudia Felser [2], Paul McClarty[3], Jeroen van den Brink [1,4,5] ✉ & Maia G. Vergniory[6,7] ✉

Altermagnets are a newly discovered class of magnetic phases that combine the spin polarization behavior of ferromagnetic band structures with the vanishing net magnetization characteristic of antiferromagnets. Initially proposed for collinear magnets, the concept has since been extended to include certain non-collinear structures. A recent development in Landau theory for collinear altermagnets incorporates spin-space symmetries, providing a robust framework for identifying this class of materials. Here, we expand on that theory to identify altermagnetic multipolar order parameters in non-collinear chiral materials. We demonstrate that the interplay between non-collinear altermagnetism and chirality allows for spatially odd multipole components, leading to non-trivial spin textures on Fermi surfaces and unexpected transport phenomena, even in the absence of SOC. This makes such chiral altermagnets fundamentally different from the well-known SOC-driven Rashba-Edelstein and spin Hall effects used in 2D spintronics. Choosing the chiral topological magnetic material $Mn_3IrSi$ as a case study, we apply toy models and first-principles calculations to predict experimental signatures, such as large spin Hall and Edelstein effects, that have not been previously observed in altermagnets. These findings pave the way for a new realm of spintronics applications based on the spin-transport properties of chiral altermagnets.

Altermagnets (AMs) are compensated magnetic phases that share features of both ferromagnets and antiferromagnets, characterized by an alternation of magnetic moments synchronized with an alternation of local multipoles around magnetic atoms[1,2].

In their original setting, altermagnets are collinearly ordered antiferromagnets that have zero net magnetization but nevertheless exhibit spin splitting of electronic bands, even in the absence of relativistic spin-orbit coupling (SOC). This leads to Fermi surfaces with non-trivial patterns of spin polarization in momentum space, for instance of $d$-wave type, that can sustain spin currents[1,2], like

ferromagnets, while being robust to stray magnetic fields, unlike their ferromagnetic counterparts. Such properties of altermagnets offer an intriguing opportunity to leverage the complementary advantages of ferro- and antiferromagnets in spintronics[1–6]. The recently developed Landau theory of collinear altermagnetism[7] provides a more general, symmetry-based definition: an antiferromagnet is considered an AM when there is neither $PT$ symmetry nor time reversal combined with a translation, and when the Néel order parameter transforms non-trivially under the point group of the lattice and leads to co-existing magnetic multipolar pseudo-primary order parameters. For more

[1]Leibniz Institute for Solid State and Materials Research, IFW Dresden, Dresden, Germany. [2]Max Planck Institute for Chemical Physics of Solids, Dresden, Germany. [3]Laboratoire Léon Brillouin, CEA, CNRS, CEA-Saclay, Université Paris-Saclay, Gif-sur-Yvette, France. [4]Würzburg-Dresden Cluster of Excellence ct.qmat, Dresden, Germany. [5]Department of Physics, TU Dresden, Dresden, Germany. [6]Département de Physique et Institut Quantique, Université de Sherbrooke, Sherbrooke, Québec, Canada. [7]Donostia International Physics Center, Donostia-San Sebastian, Spain. ✉e-mail: j.van.den.brink@ifw-dresden.de; maia.vergniory@usherbrooke.ca

general magnetic orderings, the presence of spin splitting requires, apart from the absence of $PT$ symmetry, that the spin space point group of the spin translation group does not contain the dihedral group $D_n$[8,9]. The pseudo-primary AM order parameter is directly related to the spin splitting of the band structures[1,2,10–14] and related key observables in, e.g., spin transport. While it has been widely recognized that non-collinear spin ordering can be such that the moments are fully compensated, while still generating, e.g., an anomalous Hall effect (AHE)[15–20], these observations in themselves do not provide an intrinsic connection to altermagnetism.

Here, we establish this connection by extending the scope of altermagnetic Landau theory to *non-collinear, chiral* systems. The non-collinear Néel ordering induces a secondary, symmetry-induced multipolar order parameter, which in turn provides a direct connection to physical observables[7,21]. We consider the chiral magnet Mn$_3$IrSi, belonging to the magnetic colorless space group P2$_1$3[22–24]. Our symmetry analysis shows that its experimentally observed compensated non-collinear Néel order induces multipolar secondary order parameters that are time-reversal odd: one spatially dipolar and another spatially quadrupolar. These altermagnetic order parameters impart a characteristic momentum-space spin texture. We determine the multipolar components for Mn$_3$IrSi not only from full-scale first-principles calculations, but also from a more general symmetry-appropriate magnetic model, and we establish their connection. The dipolar order parameter manifests itself through the bulk Edelstein and spin Hall effects (SHEs). What sets these effects apart in such non-collinear chiral altermagnets is that they occur in the absence of SOC, making them fundamentally different from the SOC-driven linear and nonlinear Rashba-Edelstein[25–27] and SHEs, as well as other non-trivial spin textures in chiral materials[28–30]. The key distinction is that the energy scale driving the altermagnetically induced transport mechanism is the spin splitting of electronic bands, which is tied to the local magnetic exchange energy[1,2,10,11] and is orders of magnitude larger than the relativistic SOC. Finally, we show that Mn$_3$IrSi is an excellent candidate to observe these effects, as, despite the presence of iridium, our calculations demonstrate that SOC plays a negligible role in its electronic structure and altermagnetic spin texture.

## Results

### Landau theory of chiral non-collinear AM

In order to elaborate on the nature of altermagnetism in chiral, non-collinear compensated magnets, it is useful to first introduce altermagnetism in its original context, namely in collinear, compensated magnets with an inversion center in the crystal structure. With this foundation, we will then be in a position to discuss the new physics that emerge when the inversion center and collinearity are lost.

Altermagnetism is most conveniently defined in the limit of zero SOC. This is because there is a separation of energy scales in altermagnetic systems, where the scale associated with spin splitting in the band structure is much larger than spin-orbit splittings. With this in mind, we consider a lattice with some space group $G$. In the paramagnetic phase, the magnetism is symmetric under the elements of $G$ as well as rotations in spin space and time reversal. In formulating a Landau theory for such systems, we introduce an order parameter $\mathbf{\Psi}$ corresponding to the collinear magnetic order that does not distinguish components in spin space. We may then define altermagnetism in such cases by requiring, first, that the magnetic sublattices are connected neither by an inversion nor a translation and, then, that the order parameter transforms as a *non-trivial* one-dimensional irreducible representation (IR) $\Gamma$ of the point group of the lattice. This directly restricts to a set of crystal symmetries where the two magnetic sublattices are connected by a non-symmorphic rotation or mirror operation[7]. The Landau theory is simply $F = c_2 \mathbf{\Psi} \cdot \mathbf{\Psi} + c_4 (\mathbf{\Psi} \cdot \mathbf{\Psi})^2$. With the condition on $\Gamma$, we may identify a spin-symmetric, time-odd,

spatially anisotropic order parameter of the form

$$\mathbf{O}_\Gamma = \int d^3\mathbf{r} [r_{\mu_1} \dots r_{\mu_p}] \mathbf{s}(\mathbf{r}) \qquad (1)$$

that transforms like $\Gamma$, where $\mathbf{s}(\mathbf{r})$ is the local magnetization density and $[\dots]$ denotes a symmetrization operation. As this transforms like $\Gamma$, it must enter the Landau theory through the additional term $\lambda \mathbf{\Psi} \cdot \mathbf{O}$. Therefore, $\mathbf{O}$ is a secondary (or pseudo-primary) order parameter. Interestingly, this multipolar order parameter is tied directly to the anisotropy in the spin structure of the Fermi surface. For example, in rutile crystals with chemical formula $MX_2$, where $M$ is magnetic with a sublattice Néel order parameter and point group $D_{4h}$, the relevant multipolar order parameter is $\int d^3\mathbf{r} xy\mathbf{s}(\mathbf{r})$, implying that the band structure exhibits a $d$-wave spin splitting−i.e., a rotation from $k_x$ to $k_y$ reverses the spin. This result is borne out by ab initio calculations of rutile magnets[2,11,31,32].

The Landau theory further exemplifies an important aspect of collinear altermagnets, namely that the order parameter breaks down a paramagnetic spin symmetry group to a *collinear spin group*. Spin groups enlarge the set of magnetic symmetry groups by including elements that do not transform spin and space identically[8,33–40]. For example, the rutile magnetic order includes an element with $C_2$ in spin space, reversing the moment, together with a composition of a $C_4$ and a translation in real space that swaps the magnetic sublattices. Collinear spin groups also include elements only acting on spin, including global rotations about the moment direction and $C_2T$, where $T$ is time reversal.

These concepts may be naturally generalized to non-collinear magnetic structures[7]. Again, one may define altermagnetism to correspond to magnetic order parameters at zero SOC that transform non-trivially under the point group of the lattice, but now without restricting to 1D IRs. We point out that an important difference compared to the collinear case is that bands no longer carry a global spin quantum number. Instead, there is a well-defined spin at each momentum, forming a spin texture across, for example, a Fermi surface. However, the nature of the momentum-space spin texture can still be characterized by the same kind of multipolar order parameters described in the collinear case.

We now focus on altermagnets in magnetic chiral crystals. It has been pointed out that collinear chiral crystals retain the $C_2T$ pure spin symmetry of their achiral counterparts. This operation takes $\mathbf{k} \to -\mathbf{k}$ while preserving the spin, meaning there is an effective inversion symmetry. Non-coplanar altermagnets stand out because they break this $C_2T$ symmetry, and the chirality can therefore be manifest in momentum space. Specifically, this means that the real-space part of the multipolar order parameter can be odd. We shall see how this feature is reflected in the spin texture in momentum space, and how it leads to experimental spin-transport signatures that stand apart from altermagnets studied to date.

To illustrate altermagnetism in chiral non-collinear systems, we consider Mn$_3$IrSi, which is a particularly interesting magnetic system due to its multifold topological semimetal properties[41–44]. Moreover, it belongs to the family of $\beta$-Mn type alloys: Mn$_3$TX ($T$ = Co, Rh, and Ir; $X$ = Si and Ge), all of which share the same crystal structure. Various features of these materials have been reported in the literature, including short-range magnetic ordering[45,46], an incommensurate magnetic phase[47] at high temperature, and a doping-induced magnetic phase transformation[48].

As shown in Fig. 1a, Mn$_3$IrSi has 12 Mn atoms with non-collinear local magnetic moments within the same unit cell. The local magnetic directions are denoted by red arrows. The space group of the Mn$_3$IrSi crystal is P2$_1$3 (No. 198), which belongs to the Sohncke space groups, with the magnetic atoms occupying the Wyckoff position 12$b$. It can be described by both the magnetic space group (MSG) P2$_1$3 and the spin space group (SSG) P$^{2_{100}}$2$_1^{3_{111}}$3, which, as we will see, are isomorphic. Due

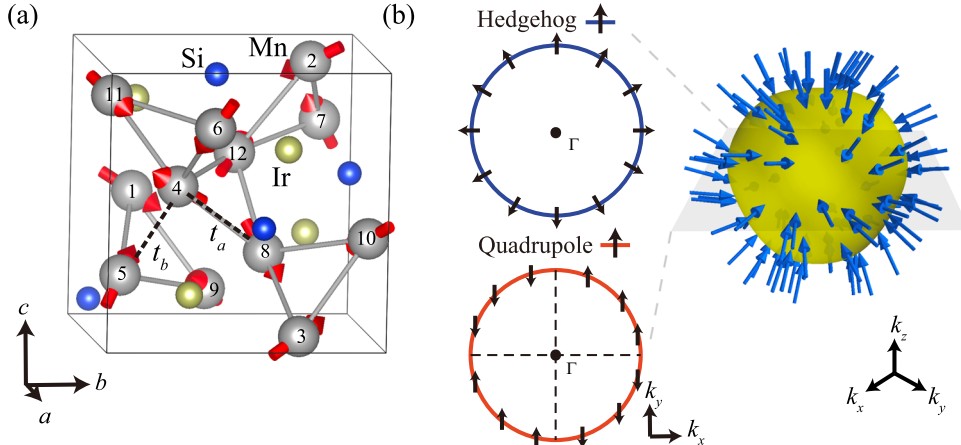

**Fig. 1 | Crystal structure of Mn₃IrSi and the momentum-space spin texture from Landau theory and a model calculation. a** The Mn₃IrSi unit cell contains 12 Mn atoms, with local magnetic moments indicated. Both the crystal and magnetic structures possess chiral symmetry, and the magnetic moments are non-collinear. The bonding between nearest and second-nearest Mn atoms is labeled by dashed black lines. **b** The dipole and quadrupole spin textures predicted in the chiral non-collinear altermagnet. They are inversion odd, $\mathbf{s}(\mathbf{k}) = -\mathbf{s}(-\mathbf{k})$, and inversion even, $\mathbf{s}(\mathbf{k}) = \mathbf{s}(-\mathbf{k})$, respectively. The three-dimensional Fermi surface is taken from the minimal-band toy model of Mn₃IrSi at $E = E_f + 0.08$ eV. The schematic plot shown in the left panel presents the dipole and quadrupole in 2D.

to the non-collinear and non-coplanar properties of its magnetic structure, the spin-only group is trivial, and the point group symmetry operations are identical in both real space and spin space. In other words, because of the non-collinear altermagnetic (AM) nature, the SSG (AM group) and the MSG of Mn₃IrSi are isomorphic.

Thus, although SOC is strictly zero in the Hamiltonian, the non-collinear magnetic ordering causes the magnetic symmetries to be equivalent to those expected in a system with finite SOC. Nevertheless, in contrast to the original collinear altermagnets, inversion symmetry is broken. However, Mn₃IrSi shares a key feature with collinear altermagnets: the absence of spin degeneracy, leading to the expectation of a spin-split band structure.

In order to understand this spin splitting from a symmetry perspective, we begin with the 12-dimensional representation based on the 12$b$ sublattice basis. This representation can be decomposed under the tetrahedral point group $T$ as: $A \oplus E \oplus 3T$. As the experimentally observed magnetic order carries zero total magnetization ($\mathbf{M} = 0$), which is also the magnetic ground state found in first-principles calculations, we can exclude a magnetic order parameter associated with the irreducible representation (IR) $A$, which corresponds to the total magnetization: $\mathbf{M} = \sum_{i \in prim} \sum_{a=1}^{12} \mathbf{S}_{ia}$, where the local moment $\mathbf{S}_{ia}$ is labeled by primitive cell $i$ and basis label $a$. We now drop the primitive lattice label as the magnetic propagation vector is $\mathbf{Q} = 0$. For the IR $E$, the magnetic structure of Mn₃IrSi is orthogonal to its components, and $\Phi_E^\alpha = 0$, where $\Phi_E^\alpha = \sum_{a=1}^{12} \phi_{E,a}^\alpha \mathbf{S}_a$, and $\alpha = 1, 2$ runs over the components of $E$. Thus, the only remaining order parameter is related to the IR $T$ (see also Ref. 39). After projecting the 12-dimensional vector onto the IR $T$, the basis of $T$ reads: $\phi_T^1 = (x, -x, -x, x, y, y, -y, -y, z, -z, z, -z)$; $\phi_T^2 = (z, -z, z, -z, x, -x, -x, x, y, y, -y, -y)$; $\phi_T^3 = (y, y, -y, -y, z, -z, z, -z, x, -x, -x, x)$. Combined with the magnetic structure: $\Phi_T^\alpha = \sum_{a=1}^{12} \phi_{T,a}^\alpha \mathbf{S}_a$. Based on the order parameter of $T$, we can write down the associated Landau theory:

$$F = c_2 \Phi_T^\alpha \cdot \Phi_{T,\alpha} + c_4 (\Phi_T^\alpha \cdot \Phi_{T,\alpha})^2 \tag{2}$$

where a sum is taken over components $\alpha$. As the total magnetization transforms as $A$, there is no direct coupling between the magnetic order parameter and $\mathbf{M}$. However, other variables related to $T$ can couple to $\Phi^\alpha$, in particular, the spatial dipole and quadrupole terms

defined as:

$$
\begin{aligned}
\mathbf{L}_{1,\alpha} &= \int d^3 r \, r_\alpha \mathbf{s}(\mathbf{r}) \\
\mathbf{L}_{2,\alpha} &= \int d^3 r \, Q_\alpha \mathbf{s}(\mathbf{r}), \quad Q_\alpha = (r_x r_y, r_x r_z, r_y r_z).
\end{aligned}
\tag{3}
$$

Based on the first term in momentum space, a hedgehog spin texture is expected, where the local spin at $\mathbf{k}$ ($\mathbf{s}(\mathbf{k})$) reverses sign under time-reversal symmetry: $\mathbf{s}(\mathbf{k}) = -\mathbf{s}(-\mathbf{k})$. For the quadrupole spatial term, this couples to a quadrupolar spin texture, which transforms differently under time-reversal symmetry: $\mathbf{s}(\mathbf{k}) = \mathbf{s}(-\mathbf{k})$. Figure 1b illustrates both the hedgehog and quadrupole-like two-dimensional spin textures, with the quadrupolar component showing zero spin at the crossing points along the $k_x$ and $k_y$ axes. The predicted spin texture, which belongs to the IR $T$, aligns with previous studies on SSGs[39].

In the original Landau theory of altermagnetism (AM)[7], an antiferromagnet is considered an altermagnet because its Néel order parameter transforms non-trivially under point group symmetries, resulting in co-existing magnetic multipolar pseudo-primary order parameters. A key characteristic of chiral crystals is the absence of inversion and mirror symmetries, meaning all improper rotational symmetries are absent. Consequently, spatial and axial vectors transform identically and can belong to the same IRs. For the chiral non-collinear altermagnet Mn₃IrSi, the non-trivial magnetic order parameter $\Phi_T^\alpha$ is established, and as a consequence, finite magnetic multipolar order parameters $\mathbf{L}_{1,\alpha}$ and $\mathbf{L}_{2,\alpha}$, along with their corresponding spin textures in momentum space, are expected on the basis of symmetry.

Finally, we consider the effect of SOC on symmetry, under the assumption that altermagnetism persists when the spin-orbit splittings are smaller than the spin splitting in the absence of SOC. When SOC is introduced, the 12 sublattices, along with the 3 local spin components for each, must be considered. This results in a 36-dimensional representation. The spin-symmetric basis in the $T$ IR now breaks into three copies of the $A$ IR, which read:

$$
\begin{aligned}
\Phi_{A1} &= -S_1^z - S_2^z + S_3^z + S_4^z - S_5^x - S_6^x + S_7^x + S_8^x - S_9^y - S_{10}^y + S_{11}^y + S_{12}^y \\
\Phi_{A2} &= -S_1^x + S_2^x + S_3^x - S_4^x - S_5^y + S_6^y + S_7^y - S_8^y - S_9^z + S_{10}^z + S_{11}^z - S_{12}^z \\
\Phi_{A3} &= -S_1^y + S_2^y - S_3^y + S_4^y - S_5^z + S_6^z - S_7^z + S_8^z - S_9^x + S_{10}^x - S_{11}^x + S_{12}^x,
\end{aligned}
\tag{4}
$$

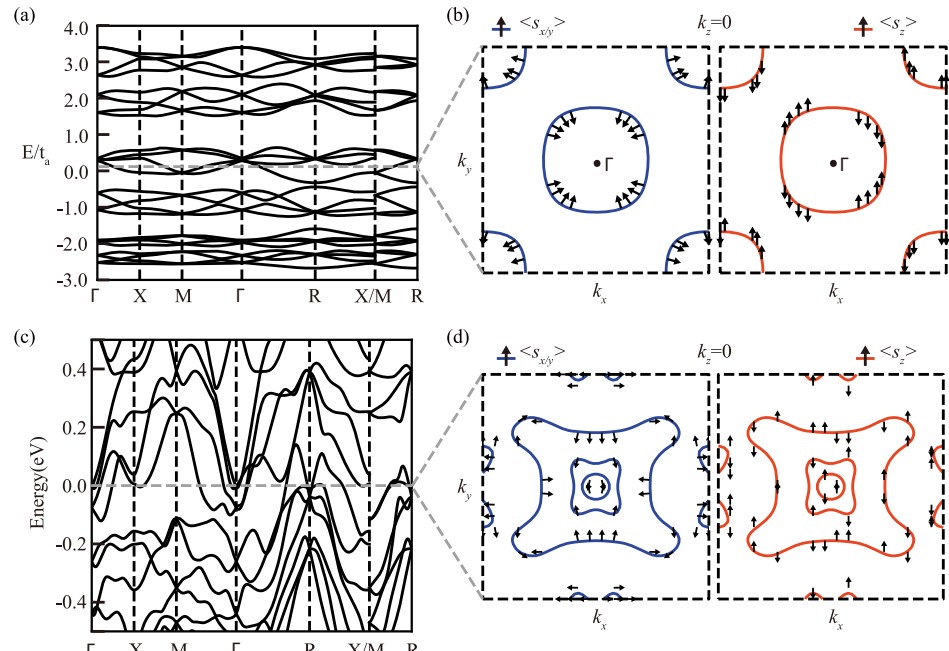

**Fig. 2 | Band structure and spin texture of Mn₃IrSi from the Kondo-lattice model and from first-principles calculations. a, c** The band structure along high-symmetry lines from the toy model and first-principles calculations, respectively. **b, d** The spin texture at $E = E_f + 0.08$ eV and at the Fermi level ($E_f$) from first-principles and toy model calculations, respectively. The left panels show the spin components $\langle s_{x/y} \rangle$, and the right panels present the spin components $\langle s_z \rangle$. All spins are normalized for clearer visualization.

whereas the magnetization transforms as $T$. Because the anti-ferromagnetic order parameter and magnetization transform differently (as $A$ and $T$, respectively), there can be no linear coupling between the two in the Landau theory. Thus, to linear order, there is no weak ferromagnetism (no small induced moment), and, as the AHE transforms like the magnetization, this too is not switched on to linear order.

## Electronic properties of Mn₃IrSi

In the previous section, we demonstrated on symmetry grounds that the non-collinear magnetically ordered phase of Mn₃IrSi must exhibit a spin texture on its Fermi surface, with dipolar and quadrupolar spatial components, even in the absence of SOC. In this section, we first develop a simple Kondo-lattice model that, with only a few parameters, accounts for the non-collinearly ordered magnetic moments. From it, the Fermi surface spin textures predicted by Landau theory are identified. Introducing SOC as a perturbation does not significantly affect the spin texture, providing a robust minimal model for the chiral non-collinear altermagnet. Next, we present the first-principles results for Mn₃IrSi, showing that its non-collinear magnetic structure stems from geometrical frustration in the absence of SOC, and that the spin texture aligns with the predictions, though it is more complex than our simplified toy model. Additionally, we demonstrate that the band structures with and without SOC are quite similar, indicating that SOC remains a small perturbation, despite the presence of the heavy element iridium. In the following section, both the toy model and the projected Wannier functions of Mn₃IrSi will be used to predict novel, robust transport phenomena that do not rely on the presence of SOC.

To capture the magnetic symmetry in Mn₃IrSi and other compounds with the same crystal structure, we propose a minimal toy model of tight-binding electrons hopping between the Mn sites on the $\beta$-Mn lattice, where magnetism is introduced via a Kondo-like coupling to fixed classical magnetic moments. These moments are arranged according to the experimentally observed non-collinear magnetic structure of Mn₃IrSi. The model is isotropic in spin space. To account for the effects of SOC, an additional symmetry-allowed spin-dependent hopping term

is included[7,49]. The Hamiltonian of the toy model is given by:

$$\hat{H} = \sum_{\alpha, \langle\langle i,j \rangle\rangle} t_{ij} a^\dagger_{i,\alpha} a_{j,\alpha} + \sum_i \mathbf{m}_i \cdot a^\dagger_{i,\alpha} \boldsymbol{\sigma}_{\alpha\beta} a_{i,\beta} + \sum_{\langle i,j \rangle} \mathbf{s}_{ij} \cdot a^\dagger_{i,\alpha} \boldsymbol{\sigma}_{\alpha\beta} a_{j,\beta} + h.c.,$$

$$(5)$$

where $t_{ij}$ represents the hopping parameters up to the second-nearest neighbors that we denote, respectively, by $t_a$ and $t_b$, $\mathbf{m}_i$ denotes the local magnetic moment strength, and $\mathbf{s}_{ij}$ is the off-site SOC strength. The Mn sublattice consists of 12 atoms, and the bases for the model are chosen to be $|i, \alpha\rangle$, where $i$ and $\alpha$ represent the sublattice and spin degrees of freedom, respectively. We limit the SOC to the nearest neighbors (**s**). Figure 2a shows the calculated band structure of the toy model along high-symmetry lines with parameters $t_b/t_a = 0.5$, $\mathbf{s}/t_a = (0.05, 0.03, 0.02)$. Without loss of generality, the local magnetic moments are chosen as $m_1/t_a = 0.3 \cdot (1.640, 2.774, -2.231)$. Figure 2b illustrates the momentum-space spin texture arising in this chiral non-collinear altermagnetic model. Only two bands cross the Fermi level, and both exhibit the hedgehog winding spin texture around the $\Gamma$ and $M$ points in $s_{x/y}$ and the quadrupole-like spin texture in the $s_z$ component.

Having established the predicted momentum-space spin texture as a generic feature of the toy model, we proceed to calculate the detailed electronic structure of Mn₃IrSi. As various mechanisms can stabilize non-collinear magnetic structures, we first identify which mechanism is at play in Mn₃IrSi. To this end, we estimate the magnetic exchange interactions in this material by constructing different collinear magnetic arrangements, computing their total energies, and mapping them onto a model of localized spins coupled by isotropic (Heisenberg) exchanges. In this way, we find that the magnetism of Mn₃IrSi can be described by a minimal model featuring three short-range antiferromagnetic exchanges, one of which—forming spin triangles adjacent to an Ir atom—is twice as large as the other two. Using classical Monte Carlo simulations, we demonstrate that this minimal model accurately reproduces the experimental non-collinear ground state, even in the absence of conduction electrons. For details of the

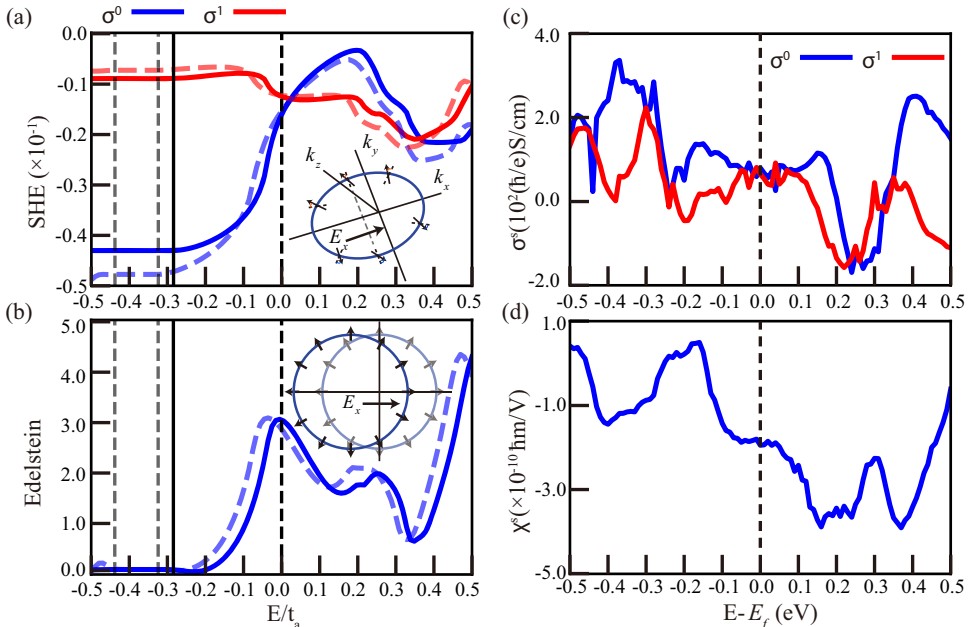

**Fig. 3 | Spin Hall and Edelstein effect conductivity tensor element** ($\sigma_{1/0} = (\sigma_{xy}^z \pm \sigma_{xz}^y)/2$) **versus energy around the Fermi level. a, c** Spin Hall effect results from the model and projected Wannier functions of $Mn_3IrSi$, respectively.

**b, d** The Edelstein effect results ($\chi^s$) with the inverse scattering time $\Gamma^2 = 10^{-4}$ ($t_a$) and (eV), respectively. The light dashed lines in (**a, b**) are from the toy model without SOC.

total-energy calculations, the minimal model, and Monte Carlo simulations, we refer the reader to Supplementary Discussions 2 and 3.

Next, we consider the band structure. As shown in Fig. 2c, the band structure along high-symmetry lines agrees well with previous reports[41]. The multifold degeneracies at the $R$ point are robust against the introduction of SOC. In the Supplementary Material, we compare the band structures shown in Supplementary Fig. 5, with and without SOC, and conclude that there is minimal difference between them.

Figure 2d shows the spin texture without SOC, revealing a hedgehog and quadrupole component in $\sigma_x$ and $\sigma_y$, and $\sigma_z$, respectively, in the $k_z = 0$ plane. The spin textures at different $k_z$ values are presented in Supplementary Fig. 1, all showing a hedgehog-like component. In addition, the spin texture in $\sigma_z$ exhibits a quadrupole-like distribution, which is even under the time-reversal operation: $s_z(\mathbf{k}) = s_z(-\mathbf{k})$. This quadrupole-like spin texture is also a consequence of the primary order parameter in collinear altermagnetism (AM)[7].

One may also understand the presence of the hedgehog and quadrupole spin textures in momentum space from the basic symmetries of chiral non-collinear AMs. Such AMs are only allowed to have rotational symmetries that relate a group of states as: $\epsilon(\mathbf{k}_i) = \epsilon(\mathbf{k}_0)$, $\mathbf{s}(\mathbf{k}_i) = R_i(\theta_i)\mathbf{s}(\mathbf{k}_0)$, $\mathbf{k}_i = R_i(\theta_i)\mathbf{k}_0$, where $R(\theta)$ is a rotation operator. From both symmetry analysis and DFT calculations of $Mn_3IrSi$, in the $k_z = 0$ plane, $\mathbf{s}(\mathbf{k}_i) = R_i(\pi/2)\mathbf{s}(\mathbf{k}_0)$, $i = x, y, z$, where $s_x$ and $s_y$ ($s_z$) transform as dipole (quadrupole), respectively. This spin-momentum locking leads to both hedgehog-like and quadrupole-like spin textures. Given the good agreement between the toy model and the realistic material calculations, we conclude that our model may be easily extended for more general applications to analyze hedgehog-like and quadrupole-like spin textures in other classes of related chiral non-collinear altermagnets.

## Spin Hall and Edelstein effects

We now show that the chiral non-collinear altermagnet discussed above exhibits both a SHE and an Edelstein effect as a consequence of the spin texture on the Fermi surface, *even in the absence of SOC*. Both phenomena arise in the presence of an externally applied electric field: the appearance of a transverse spin current signals the SHE, while the

Edelstein effect corresponds to a net magnetization. More precisely, the spin current $\mathcal{J}_j^i$ for spin component $i$ and current direction $j$ may depend on the electric field $E_k$ through $\mathcal{J}_j^i = \sum_k \sigma_{jk}^i E_k$, and the SHE corresponds to terms off-diagonal in $jk$. The Edelstein effect is a change in magnetization $\delta m_i = \chi_{ij}^s E_j$. Both phenomena are well established in various systems in the presence of SOC[50,51].

For the crystal symmetry of $Mn_3IrSi$, the spin Hall tensor $\sigma_{jk}^i$ has two independent, time-reversal even, non-vanishing components: $\sigma_{xy}^z$ and $\sigma_{xz}^y$, each transforming as the $A$ IR of the group. These, therefore, directly couple to the squared order parameter in the presence of SOC, as the order parameter also transforms like $A$. In the spin-orbit-free case, we saw that the order parameter instead transforms like $T$. As $T \otimes T$ contains $A$, there is a component that produces a SHE.

A prerequisite for a non-vanishing Edelstein effect is that the crystal must lack inversion symmetry. For the crystal structure of $Mn_3IrSi$, the Edelstein tensor has a single diagonal component $\mathbf{m} = \chi^s \mathbf{E}$ that transforms like $A$ and is time-reversal odd. This couples linearly to the order parameter in the presence of SOC. In the altermagnetic case of zero SOC, there is no linear coupling. Instead, the Edelstein effect is *cubic* in the order parameter as $T \otimes T \otimes T = 2A \oplus \dots$.

Having established that a SHE and Edelstein effect are allowed on symmetry grounds, we now show that they arise in $Mn_3IrSi$ directly from a microscopic calculation. For the spin current operator $\hat{A}_j^i = \mathcal{J}_j^i = \frac{1}{2}\{\hat{s}_i, \hat{v}_j\}$, we compute the linear spin Hall response[52,53] with a constant inverse scattering time $\Gamma$. Two components contribute to the observable $\delta \hat{A}_j^i = (\chi_{i,jk}^I + \chi_{i,jk}^{II})E_k$, where:

$$\chi_{i,jk}^I = -\frac{e\hbar}{\pi V N} \sum_{\mathbf{k},m,n} \frac{\Gamma^2 \, \mathrm{Re}\left(\langle n\mathbf{k}|\hat{A}_j^i|m\mathbf{k}\rangle \langle m\mathbf{k}|\hat{v}_k|n\mathbf{k}\rangle\right)}{[(E_f - \epsilon_{n\mathbf{k}})^2 + \Gamma^2][(E_f - \epsilon_{m\mathbf{k}})^2 + \Gamma^2]},$$

$$n = \mathrm{occ}$$
$$m = \mathrm{unocc} \tag{6}$$

$$\chi_{i,jk}^{II} = -\frac{2\hbar e}{V N} \sum_{\mathbf{k},n \neq m} \frac{\mathrm{Im}\left(\langle n\mathbf{k}|\hat{A}_j^i|m\mathbf{k}\rangle \langle m\mathbf{k}|\hat{v}_k|n\mathbf{k}\rangle\right)}{(\epsilon_{n\mathbf{k}} - \epsilon_{m\mathbf{k}})^2}.$$

Here, $e$ is the elementary charge, $\mathbf{k}$ is the Bloch wave vector, $n$, $m$ are the band indices, $\epsilon_{n,\mathbf{k}}$ is the eigenvalue, $E_f$ is the Fermi energy, $\hat{\mathbf{v}}$ is the velocity operator, $N$ is the total number of Bloch waves, and $V$ is the

volume of the unit cell. In the expression for $\chi^{II}$, the ranges of $n$ and $m$ refer to all the occupied and unoccupied bands, respectively, which is analogous to the calculation of Berry curvature. The intrinsic SHE is defined as the antisymmetric part from $\chi^{II}$ [50] that is irrelevant to scattering time and is shown in Fig. 3. The extrinsic Edelstein effect results are evaluated from $\chi^{I}$ with $\hat{A}_i = \hat{s}_i$. Other calculated transport results are presented in Supplementary Materials.

Figure 3 shows the calculated SHE (upper panels) and Edelstein effect (lower panels) for both the toy model (left-hand side) and the $Mn_3IrSi$ first-principles calculations (right-hand side). The calculations are presented at zero temperature as a function of energy relative to the Fermi level. While the magnitude and even the sign of the responses are parameter dependent, the calculations clearly reveal the presence of both a spin Hall current and an electric-field-induced magnetization, except at fine-tuned energies. We also set the relativistic effect to zero in the SHE and Edelstein effect simulations. The trends versus energy and the magnitudes are similar to those with SOC, as shown in the Supplementary Materials.

The microscopic calculations supply useful intuition by connecting the spin texture at the Fermi level to the observables. The physical mechanism is illustrated in the insets of Fig. 3a, b, where the electric field displaces the Fermi surface in reciprocal space. The hedgehog spin texture then ensures that spins at the Fermi surface no longer compensate, thus leading directly to an induced magnetization. A similar displacement causes electrons to experience an effective torque and tilt towards the $\sigma_z$ direction. A net spin current perpendicular to the electric field ($E_x$) appears due to the opposite sign of the torque between momentum $k_y > 0$ and $k_y < 0$, as indicated in the inset to panel (a).

Quantitatively, compared to other predicted spin Hall materials, $Mn_3IrSi$ has a relatively large SHE of $\sim 10^2 (\hbar/e)$ S/cm [54]. This is an order of magnitude larger than SOC-induced intrinsic values calculated for, e.g., nonmagnetic GeAs, AlAs, or Ge [55], and comparable to the predicted SHE in collinear antiferromagnets [56,57]. The value of the Edelstein effect caused by the chiral, non-collinear altermagnetism in $Mn_3IrSi$ is of the same order of magnitude as that of certain non-coplanar magnets [58].

## Discussion

Using group theory and Landau theory, we have predicted the existence of non-collinear chiral altermagnets and their distinctive electronic and transport properties. Compared to collinear altermagnets, non-collinear systems exhibit a more intricate momentum-space spin texture, extending the classification scheme for collinear systems. On theoretical grounds, we have established the presence of a unique spin-momentum locking mechanism that arises in the absence of SOC, as a direct consequence of chiral altermagnetism in $Mn_3IrSi$. This generalizes altermagnetism from its original context in collinear magnets—where entire electronic bands can be labeled by a common spin quantum number—to systems where the bands possess a local spin degree of freedom that is globally constrained to form a momentum-space texture governed by symmetry. This spin texture has direct physical implications. Due to the stricter symmetry constraints in non-chiral altermagnets, spatially odd multipole components are not allowed in either collinear or non-collinear magnetic structures [7]. Thus, chirality emerges as one of the necessary conditions for the hedgehog spin texture in AMs. As a consequence, as exemplified in $Mn_3IrSi$, large spin Hall ($\sim 10^2 (\hbar/e)$ S/cm) and Edelstein (approximately $-2 \times 10^{-10} \hbar$ m/V) effects coexist and have been calculated, showing very weak SOC dependence. We foresee multifunctional applications of chiral non-collinear AMs in spintronics and suggest additional candidates from the $Mn_3IrSi$ family: $Mn_3IrGe$ [59], $Mn_3CoGe$ [48], $Mn_3RhGe$ [47], $Mn_3IrGe$ [59]; and other chiral non-collinear AMs: $ScMnO_3$ [60], $BaCuTe_2O_6$ [61,62], $YMnO_3$ [63], $Ho_2Ge_2O_7$ [64], $Er_2Ge_2O_7$ [65].

## Methods

Spin-polarized scalar-relativistic total-energy calculations were performed using the generalized gradient approximation (GGA) [66] as implemented in the full-potential code FPLO [67] version 22 on a $k$-grid of $8 \times 8 \times 8$ points. Exchange interactions were obtained by mapping the 181 GGA total energies onto an $S = \frac{3}{2}$ Heisenberg model, and by a least-squares solution of the latter. Classical Monte Carlo simulations of the Heisenberg model were performed using ALPS [68,69] version 2.3.0 on a finite lattice of $4 \times 4 \times 4$ cells (768 spins) with periodic boundary conditions; we employed local updates and used 5,000,000 (500,000) sweeps for measurement (thermalization). Band structure calculations were performed using VASP [70], employing the projector augmented wave method [71]. The Brillouin zone was sampled on a $7 \times 7 \times 7$ $k$-point grid centered at the Gamma point. The energy cutoff for the plane wave basis was set to 550 eV. The Hubbard term was introduced with a value of 3.0 eV for the $d$ orbitals of the Mn atoms within the DFT+$U$ framework to account for electron-electron correlations. The Wannier-based Hamiltonian was symmetrized based on the maximally localized Wannier functions generated by the WANNIER90 interface [72]. The projectors were the $d$ orbitals of Mn, with the fitted region spanning from $-2$ to $2$ eV. The magnetic moments within the self-consistent non-collinear ground state, with and without SOC, are respectively: $|\mathbf{m}| = 3.91 \pm 10^{-2} \mu_B$ and $\mathbf{m} = (1.640,\ 2.774,\ -2.231) \pm 10^{-3} \mu_B$ on the Mn atom located at $(0.1195, 0.2031, 0.4573)$ in fractional coordinates of the unit cell.

## Data availability

The supporting data for this study are available from the corresponding authors upon reasonable request.

## Code availability

The computation code in terms of transport properties in this study is available from the corresponding authors upon reasonable request.

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

## Acknowledgements

M.G.V. acknowledges fruitful discussions with Fernando de Juan. M.G.V. thanks support to the Deutsche Forschungsgemeinschaft (DFG, German Research Foundation) GA 3314/1-1—FOR 5249 (QUAST), the Spanish Ministerio de Ciencia e Innovación (PID2022-142008NB-I00), the Canada Excellence Research Chairs Program for Topological Quantum Matter and funding from the IKUR Strategy under the collaboration agreement between Ikerbasque Foundation and DIPC on behalf of the Department of Education of the Basque Government. M.H. and O.J. thank Ulrike Nitzsche for technical assistance. M.H. thanks the support from the Alexander von Humboldt Foundation. We acknowledge financial support by the Deutsche Forschungsgemeinschaft (DFG, German Research Foundation), through SFB 1143 (Project ID 247310070), project A05, Project No. 465000489, and the Würzburg-Dresden Cluster of Excellence on Complexity and Topology in Quantum Matter, ct.qmat (EXC 2147, Project ID 390858490).

## Author contributions

M.V. and J.V.D.B. proposed the project. M.H., P.M., O.J., M.V. and J.V.D.B. conceived the project. M.H. did first-principles and transport calculations and O.J. conducted the Monte Carlo and magnetic exchange calculations. P.M. did the Landau theory. M.H., P.M., O.J., J.V.D.B. and M.V. wrote the paper. M.H., O.J., P.M., M.V. and J.V.D.B. contributed to the scientific discussions. M.H., O.J., C.F., P.M., M.V. and J.V.D.B. participated and commented on the paper.

## Funding

## Competing interests

The authors declare no competing interests.
