## [Peer Review File · Nature Communications]

Spin Hall and Edelstein Effects in Chiral Non-collinear Altermagnets

Corresponding Author: Dr Mengli Hu

Version 0:

Reviewer comments:

Reviewer #1

(Remarks to the Author)

By using the effective model and the Landau theory, the authors were trying to expand the altermagnet to non-collinear magnetic systems. As the authors stated in the manuscripts, the concept of altermagnet is defined in collinear magnetic systems in previous works and was used in a collinear space group to understand such effects. In this work, the authors argued that the concept of altermagnet could also be defined via the Landau theory and they expanded this concept to noncollinear magnetic systems. Furthermore, the authors proposed Mn₃IrSi as the material candidate to support the non-collinear altermagnet. Somehow, I think the authors are trying to provide something new to this community and deepen the understanding of the altermagnet. While I still have some questions about the definition and calculated results before, the publication. The detailed comments can be found following.

- 1, For the collinear altermagnet, the collinear space group can be applied to such a system to classify and understand the altermagnetic properties. Herein, for the non-collinear altermagnet, do you have a space group theory to classify the whole system?
- 2, Without SOC, why is the non-collinear magnetic structure the magnetic ground state? Did the authors have some general arguments for stabilizing the non-collinear magnetic structure in a system without SOC?
- 3, For Mn₃IrSi, from the view of material science, the discussion shown in this paper is quite weak. Is the non-collinear magnetic structure shown in the paper as the ground state? Why? I expect the authors to write down a magnetic coupling model to understand the origin of non-collinear magnetic structure.
- 4, The Band structures from the Kondo lattice model and first principles calculations are totally different. I do not think the results from the Kondo lattice model can represent the electronic and transport properties of Mn₃IrSi.

Reviewer #3

(Remarks to the Author)

Please find my report attached.

Reviewer #4

(Remarks to the Author)

Version 1:

Reviewer comments:

Reviewer #1

(Remarks to the Author)

I appreciate the authors considering my comment seriously, especially to build a magnetic model to discuss the origin of non-collinear features in Mn₃IrSi. I think the authors answered my questions and comments properly. I am happy to support the publication of this work in Nature Communications.

Reviewer #3

(Remarks to the Author)

I am satisfied with the authors' response. The revised manuscript is now clearer than before. I recommend the publication of the revised manuscript at Nature Communications after the authors consider the following suggestions:

[1] ϕ_{T^1} , ϕ_{T^2} , ϕ_{T^3} at the bottom of p. 7 and at the beginning of p. 8 seem to assume a specific labeling of the 12 magnetic atoms ($a=1, 2, 3, \dots, 12$) in a unit cell. I suggest that the authors provide these labels in Fig. 1(a) to help readers understand the physical meaning of the IR T .

[2] In Eq. (4), the symbol ϕ is used in ϕ_{A1} , ϕ_{A2} , ϕ_{A3} to denote the IR A . However, in the earlier part of the manuscript, the symbol Φ (uppercase) is used to denote IRs and the symbol ϕ (lowercase) to denote the basis of IRs. For notational consistency, I recommend that the authors use Φ (uppercase) to denote the IR A in Eq. (4).

[3] 1st line in p.12: It was stated that α denotes the sublattice. However, the third term in the Hamiltonian H [Eq. (5)] seems to imply that α is the spin index. I request the authors to clarify this point.

[4] 4th line in the second paragraph at p. 13: Does ϵ_0 amount to $\epsilon(k_0)$? If so, I suggest that the authors replace ϵ_0 with $\epsilon(k_0)$.

[5] 2nd line from the bottom of p.13: It was stated that "For the crystal symmetry of Mn₃IrSi, the spin Hall tensor $\sigma^i_{\{jk\}}$ has two independent time-reversal even, non-vanishing components: $\sigma^z_{\{xy\}}$ and $\sigma^y_{\{xz\}}$, each ..." I was confused by this statement. If I ignore the magnetic moment arrangement and focus on the crystal symmetry, the 90-degree rotation around the x-axis followed by a proper fractional translation is a nonsymmorphic symmetry of the crystal structure and forces the two components to have the same magnitude with opposite signs. So I guess what the authors intended is the magnetic symmetry instead of the crystal symmetry.

[6] p.14: It was stated in the paragraph right below Eq. (6) that only the second expression in Eq. (6) is relevant for the intrinsic spin Hall effect and only the first expression is relevant for the Edelstein effect. On the other hand, the description right above Eq. (6) implies that both expressions contribute to both the spin Hall effect and the Edelstein effect. I am afraid that this inconsistency may cause confusion to readers. I suggest that the authors revise this part of the manuscript to avoid possible confusion.

Reviewer #4

(Remarks to the Author)

This manuscript attempts to extend the Landau theory description of altermagnetism, which has been discussed in collinear systems [6], to non-collinear chiral altermagnets. Through the extension, the authors explore how the noncollinearity may be related to the non-relativistic spin splitting and magnetic multipoles. Furthermore, as an application of this non-relativistic spin splitting, the authors demonstrate the Edelstein effect and the spin Hall effect.

The manuscript deals with an interesting subject at the frontier of the altermagnetism study, and I expect this work to attract the community's attention. However, I have some concerns and questions about the manuscript, which need to be addressed to clarify the novelty and general applicability of the manuscript. I look forward to the authors' responses and revisions.

[1] Multipoles

In p.3, it is stated that “non-collinear Néel order induces one monopolar and three quadrupolar altermagnetic secondary order parameters.” However, it is not clear to me what they mean by the monopolar altermagnetic secondary order parameters. A possibly related quantity is given at the bottom of p.7, where the spatial dipole $\mathbf{L}_{1,\alpha}$ is presented. Does $\mathbf{L}_{1,\alpha}$ amount to the monopolar altermagnetic secondary order parameter? If so, it is not clear to me why the authors called the spatial *dipole* $\mathbf{L}_{1,\alpha}$ *monopolar* altermagnetic secondary order parameter.

I also have a question on the number of multipoles. In p.3, the authors stated that there are three monopolar (or dipolar?) and three quadrupolar altermagnetic secondary order parameters. However, the equation for the spatial dipole $\mathbf{L}_{1,\alpha}$ in 7 allows three components ($\alpha = x, y, z$). Thus, if $\mathbf{L}_{1,\alpha}$ amounts to the monopolar altermagnetic secondary order parameter, the number of $\mathbf{L}_{1,\alpha}$ does not match the number of monopolar altermagnetic secondary order parameters stated in p.3. The number of quadrupolar altermagnetic secondary order parameters is mysterious to me. In p.3, the authors stated that there are three quadrupolar altermagnetic secondary order parameters. However, Fig. 2 shows only one quadrupolar distribution of $\langle S_z \rangle$ in the momentum space. If the quadrupolar altermagnetic secondary order parameters amount to the quadrupole distribution of $\langle S_z \rangle$ in the momentum space, I don't understand why the number of the quadrupolar altermagnetic secondary order parameters differs from the number of the quadrupolar distribution of $\langle S_z \rangle$ in the momentum space.

The multipole order parameters $\mathbf{L}_{1,\alpha}$ and $\mathbf{L}_{2,\alpha}$ should couple to the primary order parameter Φ^α . What are their coupling structures?

[2] Spatial dipole vs. hedgehog spin texture

In p.7, it is stated that a hedgehog spin texture is expected on the basis of the spatial dipole. $L_{1,\alpha}$. I understand that the dipole $L_{1,\alpha}$ may imply the relation $\mathbf{s}(\mathbf{k}) = -\mathbf{s}(-\mathbf{k})$. However, I don't understand how this relation leads to the hedgehog spin texture. Although the hedgehog spin texture satisfies the relation $\mathbf{s}(\mathbf{k}) = -\mathbf{s}(-\mathbf{k})$, the relation does not necessarily imply the hedgehog spin texture. For instance, the p -wave spin splitting [8] also satisfies the relation but has a qualitatively different spin texture from the hedgehog spin texture. So, it is unclear whether the symmetry analysis guarantees the hedgehog spin texture. If so, I wonder what the main difference is in terms of the symmetry analysis that distinguishes p -wave altermagnets from Mn_3IrSi with the hedgehog spin texture. A related question is how broadly the results of this manuscript apply beyond Mn_3IrSi . Does the presence of a non-collinear chiral altermagnet always imply the existence of a hedgehog spin texture? If not, what key conditions determine whether a system falls into this classification? Providing a more generalized argument will strengthen the impact of the work.

[3] Chirality

The authors emphasize both noncollinearity and chirality. From the manuscript, I can see that the noncollinearity makes a qualitative difference from the collinearity. However, it is not clear to me what role the chirality plays. In p. 6, it is stated that “Non-collinear altermagnets stand out because they break this C_2T and the chirality can therefore be manifest in momentum space. Specifically, this means that the real space part of the multipolar order parameter can be odd.” So, this statement seems to imply that the chirality is necessary to violate the constraint $\mathbf{s}(\mathbf{k}) = \mathbf{s}(-\mathbf{k})$. However, this constraint can be violated once the inversion symmetry is broken. Although the inversion symmetry breaking is a necessary condition for the chirality, it is not a sufficient condition since the chirality requires the breaking of additional symmetries. If the inversion symmetry breaking alone is sufficient to violate the constraint $\mathbf{s}(\mathbf{k}) = \mathbf{s}(-\mathbf{k})$, it is more appropriate to say that the noncentrosymmetry is needed instead of the chirality for the real space part of the multipolar order parameter to have odd components.

By the way, it is stated in p.15 that “Due to the stricter symmetry constraints in non-chiral altermagnets (AM), spatially odd multipole components are not allowed in either collinear or non-collinear magnetic structures [6]”. So, this statement indicates that the chirality (instead of the noncentrosymmetry) is necessary for the odd components of the multipolar order parameters. Ref. [6] appears to be the source of this statement. However, I could not find any related discussion in Ref. [6]. Therefore, I request the authors to clarify the role of the chirality for the odd components.

[4] SOC dependence

A few lines above Sec. VI, it is stated that “large spin Hall ... and Edelstein ... effects co-exist and have been calculated, showing very weak SOC dependence.” However, I don’t find any related discussion on the weak SOC dependence in Sec. IV SPIN HALL AND EDELSTEIN EFFECTS. The only place where the SOC dependence appears is the caption of Fig. 3, which states that the light dashed lines in Fig. 3(a) and (b) are obtained for the model Hamiltonian without SOC. To make the claim of the weak SOC dependence convincing, I recommend the spin Hall effect and the Edelstein effect in Mn₃IrSi to be calculated through the first-principles calculations and explicitly compared for the cases with and without SOC.

[5] Relationship between odd-parity multipoles and odd-parity spin splitting (hedgehog spin texture)

In collinear *d*-wave altermagnets, real-space magnetic octupoles ($r_m r_n S_q$) are directly related to the \mathbf{k} -space spin-even parity splitting ($k_m k_n S_q$), which is natural considering that they transform the same way under the time-reversal and also under the space inversion. However, the connection between odd-parity multipoles ($L_{1,\alpha}$ for example) and odd-parity spin splitting (hedgehog spin texture) is not obvious since these two quantities transform differently under the time-reversal (T) and under the inversion combined time-reversal (PT). For example, in Mn₃IrSi, flipping all spin directions (applying T) invert $L_{1,\alpha}$, but does not invert the hedgehog spin texture. Are these two quantities directly related, or do they exhibit a more subtle dependence? A detailed discussion clarifying this point would be helpful.

I have additional comments and questions, which are relatively minor compared to those given above.

[6] In p.5, it is stated that “The Landau theory further exemplifies an important aspect of collinear altermagnets namely that the order parameter breaks down a paramagnetic spin symmetry group to a collinear spin group.” In this statement, does the “order parameter” refer to the primary order parameter Ψ or the secondary order parameter O_Γ ?

[7] In p.6, it is stated that “Non-collinear altermagnets stand out because they break this C_2T and ...” I understand this statement for noncoplanar magnets. However, for coplanar magnets, isn’t C_2T preserved?

[8] In p.7, the definition of \mathbf{M} is given: $\mathbf{M} = \sum_{i \in \text{prim}} \sum_{a=1}^{12} \mathbf{S}_{ia}$. I don’t understand index *i* in this definition. What does it refer to?

[9] In p.7, it is stated that “ $\forall \alpha, \Phi^\alpha = 0$ ” for the IR *E*, where $\Phi^\alpha = \sum_{a=1}^{12} \phi_{E,a}^\alpha \mathbf{S}_a$. I don’t understand this statement. What is meant by “ $\forall \alpha, \Phi^\alpha = 0$ ” when there is only one IR *E*?

[10] In p.7, the expression for the Landau theory in p.7 reads, “ $F = c_2 \Phi^\alpha \cdot \Phi_\alpha + c_4 (\Phi^\alpha \cdot \Phi_\alpha)^2$ ”. In this expression, is the summation over α assumed? If so, should the coefficients c_2 and c_4 be independent of α ?

[11] In p.7, the formula for the spatial dipole $L_{1,\alpha}$ reads $L_{1,\alpha} = \int d^3r r_\alpha \mathbf{m}(\mathbf{r})$. How does $\mathbf{m}(\mathbf{r})$ compare with $\mathbf{s}(\mathbf{r})$ in the formula for \mathbf{O}_Γ in p.5? Are they the same? If so, I recommend the authors use the same variables to avoid possible confusion.

[12] In p.8, it is stated that “The spin-symmetric basis in the T IR now breaks into three copies of the A IR.” Please clarify this statement. Considering that there are three T IRs in the absence of SOC, does it mean that each T IR in the absence of SOC breaks into three copies of A IR in the presence of SOC?

[13] Equation (2) in p.11 contains the hopping coefficient $t_{i,j}$. How is it related to $(t_i, i = a, b)$ given five lines below Eq. (2)?

[14] In the 2nd paragraph in Sec. IV, it is stated that “These therefore directly couple to the squared order parameter in the presence of SOC as the order parameter also transforms like A . Does the order parameter refer to the primary order parameter or the secondary order parameter? Similarly, in the 3rd paragraph in Sec. IV, it is stated that “Instead, the Edelstein effect is *cubic* in the order parameter as ...” Does the order parameter refer to the primary order parameter or the secondary order parameter?

**Reply to Referees: “Spin Hall and Edelstein Effects in Novel Chiral
Non-collinear Altermagnets”**

I. REPLY TO REFEREE 1

By using the effective model and the Landau theory, the authors were trying to expand the altermagnet to non-collinear magnetic systems. As the authors stated in the manuscripts, the concept of altermagnet is defined in collinear magnetic systems in previous works and was used in a collinear space group to understand such effects. In this work, the authors argued that the concept of altermagnet could also be defined via the Landau theory and they expanded this concept to noncollinear magnetic systems. Furthermore, the authors proposed Mn_3IrSi as the material candidate to support the non-collinear altermagnet. Somehow, I think the authors are trying to provide something new to this community and deepen the understanding of the altermagnet. While I still have some questions about the definition and calculated results before, the publication. The detailed comments can be found following.

1, For the collinear altermagnet, the collinear space group can be applied to such a system to classify and understand the altermagnetic properties. Herein, for the non-collinear altermagnet, do you have a space group theory to classify the whole system?

We thank the referee for the question. In terms of the appropriate group to describe the properties of non-collinear AM, the spin space group definition is general and powerful enough to describe both collinear and non-collinear systems. For the non-collinear altermagnet, the spin-only group is, in some sense, simpler since it is trivial (there is no continuous rotational symmetry in spin space). The classification, within the scope of our work, refers to features that are constrained by these spin-space symmetries. These include the spin texture (as with the hedgehog in Mn_3IrSi) and the response of the system (such as the Spin Hall and Edelstein effects). In the chiral non-collinear AM system, we are, in a very general way, demonstrating that the hedgehog spin texture is correlated with the presence of electric-field-induced phenomena arising from both the non-collinear magnetic structure and the chiral symmetry. These two conditions are equally important.

The purpose of this work is to illustrate, in a specific instance, new phenomena arising in the zero spin-orbit coupling limit for non-coplanar magnetic structures, thereby generalizing the concepts of altermagnetism beyond the collinear limit and connecting them to measurable quantities. A complete classification of such non-collinear systems is an interesting issue that is beyond the scope of this work.

2, Without SOC, why is the non-collinear magnetic structure the magnetic ground state? Did the authors have some general arguments for stabilizing the non-collinear magnetic structure in a system without SOC?

We thank the referee for these questions. We agree that non-collinear magnetism can have different origins, such as geometrical frustration, competing anisotropic interactions, and single-ion anisotropies. The latter can be excluded: in single-crystal measurements, the magnetic susceptibility of Mn_3IrSi shows some directional dependence in both the paramagnetic and ordered states [1], but this anisotropy reaches at most 5%. By the same argument, the role of anisotropic interactions is likely minor. We can therefore expect that the non-collinearity stems from geometrical frustration, i.e., the competition of isotropic exchange interactions. Indeed, the crystal structure of Mn_3IrSi features triangular loops, yet there are many possible short-range Mn..Mn connections, and it is a priori unclear which of them are relevant. Direct band-structure insights into this question (for instance, by applying the magnetic force theorem) are impeded by the complex band structure: our attempts to describe the GGA valence band using a Wannier model of Mn $3d$ states were unsuccessful.

In the absence of information on relevant couplings, we assume that the leading magnetic exchanges are short-range and restrict our analysis to those with $d_{\text{Mn..Mn}} \leq 5 \text{ \AA}$. There are 12 such exchanges (Tab. R1), where the three shortest exchanges J_1 , J_2 , and J_3 together form a trillium lattice. To estimate these exchanges, we calculate the GGA total energies of various collinear magnetic arrangements and map them onto a classical $S = \frac{3}{2}$ Heisenberg model. To keep the problem computationally tractable, we refrain from doubling the unit cell (as a result, some of the exchanges can be estimated only as a sum), and instead make all 12 Mn atoms inequivalent by removing all symmetry elements (space group $P1$). In this cell, the total number of different magnetic configurations is $2^{(12-1)} = 2048$ (the minus one in the exponent is due to the global spin inversion), yet assuming the magnetism is fully described by the twelve short-range exchanges, only 184 of these configurations are inequivalent. Three configurations failed to converge; the remaining 181 total energies were used to parameterize the $S = \frac{3}{2}$ Heisenberg model. Parameterization is done by a least-squares solution of the redundant linear problem, which gives the following solution:

TABLE R1: Short-range ($d_{\text{Mn..Mn}} \leq 5 \text{ \AA}$) magnetic exchanges in the crystal structure of Mn_3IrSi . Interatomic distances correspond to the 5 K structure from Ref. 2.

exchange	$d_{\text{Mn..Mn}}, \text{ \AA}$	multiplicity	exchange	$d_{\text{Mn..Mn}}, \text{ \AA}$	multiplicity
J_1	2.6888	24	J_7	4.5483	24
J_2	2.7206	24	J_8	4.5677	24
J_3	2.7658	24	J_9	4.6360	24
J_4	3.3450	24	J_{10}	4.6962	24
J_5	3.9885	24	J_{11}	4.6976	24
J_6	4.4663	24	J_{12}	4.8354	24

$$\begin{aligned}
 J_1 &= 14.0(5) \text{ meV} & J_4 + J_{10} &= -1.2(3) \text{ meV} \\
 J_2 + J_8 &= 23.0(4) \text{ meV} & J_5 + J_{11} + J_{12} &= -1.9(5) \text{ meV} \\
 J_3 + J_9 &= 11.8(4) \text{ meV} & J_6 &= 1.2(3) \text{ meV} \\
 & & J_7 &= 1.4(3) \text{ meV}
 \end{aligned} \tag{R1}$$

The error bars and the fit quality (Fig. R1) are reasonable: Mn_3IrSi is a metal and the local moments on Mn depend on the configuration – averaging over all Mn sites and all computed configurations yields $3.22(10) \mu_{\text{B}}$. The good agreement with the GGA energies *a posteriori* confirms our assumption that longer-range exchanges are small. Following this logic, we can further assume $J_2 \gg |J_9|$ and $J_3 \gg |J_{11}|$. This leaves us with a rather simple minimal model comprising three relevant antiferromagnetic exchanges J_1 , J_2 , and J_3 whose relative strength is approximately 1:2:1. All further exchanges are at least about 5 times smaller than these leading terms.

Next, we compute the magnetic susceptibility of the classical J_1 - J_2 - J_3 Heisenberg model by performing classical Monte Carlo simulations on $4 \times 4 \times 4$ finite lattices (each unit cell comprises 12 spins) with periodic boundary conditions, assuming $J_1 : J_2 : J_3 = 1 : 2 : 1$ (the minimal model of Mn_3IrSi), and compare it to the regular trillium lattice ($J_1 = J_2 = J_3$). Both curves (Fig. R2) show a distinct kink that indicates long-range magnetic ordering. It is crucial to note that the Heisenberg model lacks contributions from itinerant electrons. In fact, the coexistence of itinerant and localized d electrons in Mn_3IrSi closely resembles the double-exchange model relevant for manganites $(\text{La}, \text{Sr})\text{MnO}_3$, where doping drastically enhances the ordering temperature [3]. In our model, we obtain a transition temperature of $0.14\bar{J}$ ($0.16\bar{J}$) for $J_1 : J_2 : J_3 = 1 : 2 : 1$ ($J_1 = J_2 = J_3$), corresponding to $S^2\bar{J}/k_{\text{B}} \simeq 60 \text{ K}$ and hence – in accord with expectations – significantly underestimating the experimental ordering temperature of 225 K. Still, we believe that the nature of the magnetically ordered state is captured by a model of localized spins. To analyze the structure

FIG. R1: Least-squares solution [Eq. R1] of the classical $S = \frac{3}{2}$ Heisenberg model parameterized with GGA energies of different collinear magnetic configurations of Mn_3IrSi . Left: differences between the total energies E_{fit} obtained from the least-squares solution and the corresponding GGA total energies E_{GGA} (the diagonal, which marks zero difference, is a guide to the eye). Right: histogram of the differences $E_{\text{fit}} - E_{\text{GGA}}$.

of the ground state, we compute the temperature dependence of spin correlations corresponding to J_1 , J_2 , and J_3 bonds (Fig. R3). At low temperatures, all three curves approach $\langle \mathbf{S}_i \cdot \mathbf{S}_j \rangle = -0.5$, which corresponds to a 120° angle between the magnetic moments. This directly proves that the non-collinear magnetic structure in Mn_3IrSi stems primarily from magnetic frustration; the latter is induced by antiferromagnetic isotropic interactions between localized moments on a trillium lattice.

FIG. R2: Magnetic susceptibility of the classical trillium-lattice Heisenberg model as a function of temperature.

FIG. R3: Spin correlations in the classical trillium-lattice Heisenberg model as a function of temperature.

3, For Mn₃IrSi, from the view of material science, the discussion shown in this paper is quite weak. Is the non-collinear magnetic structure shown in the paper as the ground state? Why? I expect the authors to write down a magnetic coupling model to understand the origin of non-collinear magnetic structure.

To answer this question, we continue our microscopic analysis. After demonstrating that the classical Heisenberg model with the exchange parameters relevant for Mn₃IrSi has a non-collinear magnetic ground state, we are now in a position to compare our results with experiment. To this end, we inspect the angles between the magnetic moments in the ordered state. The magnetic structure of Mn₃IrSi has been measured at 10 and 5 K, as reported in Refs. 4 and 2, respectively. Note that slight differences in the atomic coordinates affect the interatomic separations: while the strongest exchange J_2 is the second shortest Mn..Mn separation (2.721 Å) in Ref. 2, it corresponds to the shortest separation (2.678 Å) in Ref. 4. To avoid confusion, we summarize the relevant crystallographic data in Tab. R2. We further note that despite the similarity of the bond lengths and the associated difficulties in the naming convention, the local crystalline environments of the spin triangles are quite different (Fig. R4): The triangle formed by J_1 is enclosed in a bipyramid whose apices are Si and Mn atoms, while triangles formed by J_2 and J_3 are bases of pyramids with, respectively, an Ir or Si atom at the apex. The strongest exchange J_2 is thus associated with Mn₃Ir pyramids.

Next, we compare the angles between the magnetic moments in both magnetic structures (Tab. R3). In the structure from Ref. 4, the angle between the moments on J_2 bonds is notably

TABLE R2: Relevant structural parameters and interatomic distances from Refs. 4 and 2.

parameter	Eriksson et al. , Phys. Rev. B 69 , 054422 (2004)	Hall et al. , Phys. Rev. Mater. 7 , 114402 (2023)
T , K	10.	5.
a , Å	6.48790	6.49081
Mn x/a	0.11950	0.1187
Mn y/b	0.20310	0.2074
Mn z/c	0.45730	0.4544
$d(J_1)$, Å	2.699	2.689
$d(J_2)$, Å	2.678	2.721
$d(J_3)$, Å	2.796	2.766

FIG. R4: Crystalline environments of spin triangles formed by J_1 (left), J_2 (centre), and J_3 (right) bonds in the crystal structure of Mn_3IrSi .

close to 120° . The angles on J_1 and J_3 are significantly smaller, which agrees very well with our estimates. In the magnetic structure from Ref. 2, this tendency is less pronounced: the angles on J_2 and J_3 are similar and somewhat smaller than 120° . A more conclusive analysis is impeded by sizable standard deviations, which amount to several degrees.

In summary, the ground state of our spin model agrees with experiment. The leading exchange J_2 in our model corresponds to a nearly perfect 120° arrangement of Mn moments in the magnetic structure of Ref. 4.

Changes made: We added this information to Sec. III clarifying the origin of the non-collinear ground state in Mn_3IrSi and refer the interested reader to Supplemental Discussions 2 and 3, where the respective first-principles calculations and Monte Carlo simulations are detailed.

TABLE R3: Angles (in $^\circ$) between local magnetic moments on Mn atoms. Standard deviations (computed from the standard deviations of $[m_x, m_y, m_z]$) are given in brackets.

bond	$\angle(m_{\text{Mn}_i}, m_{\text{Mn}_j})$	
	Eriksson et al. , Phys. Rev. B 69, 054422 (2004)	Hall et al. , Phys. Rev. Mater. 7, 114402 (2023)
J_1	100(3)	104(2)
J_2	119.4(5)	111(3)
J_3	103(2)	111(4)

4, The Band structures from the Kondo lattice model and first principles calculations are totally different. I do not think the results from the Kondo lattice model can represent the electronic and transport properties of Mn₃IrSi.

The referee is correct that many details differ between the full band structure calculations and the toy model. The purpose of the toy model is to illustrate the universal features of any band structure at zero spin-orbit coupling arising from the interplay between the magnetic structure and the itinerant electrons (where the magnetism may originate from the itinerant electrons themselves). By including the toy model calculation, we aim to demonstrate that the details of the microscopic calculation do not affect the bulk properties and global spin texture on the Fermi surface. This further reinforces the point that the detailed origin of the magnetic structure, while interesting, is beyond the scope of this study. Any mechanism leading to the observed magnetic structure will result in similar features.

II. REPLY TO REFEREE 2

This manuscript attempts to extend the Landau theory description of altermagnetism, which has been discussed in collinear systems [6], to non-collinear chiral altermagnets. Through the extension, the authors explore how the noncollinearity may be related to the non-relativistic spin splitting and magnetic multipoles. Furthermore, as an application of this non-relativistic spin splitting, the authors demonstrate the Edelstein effect and the spin Hall effect. The manuscript deals with an interesting subject at the frontier of the altermagnetism study, and I expect this work to attract the community's attention. However, I have some concerns and questions about the manuscript, which need to be addressed to clarify the novelty and general

applicability of the manuscript. I look forward to the authors’ responses and revisions

We thank the referee for a careful reading of our manuscript and for their perceptive questions. The following addresses all of the referee’s questions, which have also resulted in various clarifications and improvements to the manuscript.

A. Questions about Multipoles

In p. 3, it is stated that “non-collinear Néel order induces one monopolar and three quadrupolar altermagnetic secondary order parameters.” However, it is not clear to me what they mean by the monopolar altermagnetic secondary order parameters. A possibly related quantity is given at the bottom of p. 7, where the spatial dipole $L_{1,\alpha}$ is presented. Does $L_{1,\alpha}$ amount to the monopolar altermagnetic secondary order parameter? If so, it is not clear to me why the authors called the spatial dipole $L_{1,\alpha}$ the monopolar altermagnetic secondary order parameter.

We thank the referee for highlighting some points that deserve clarification in the text. We find that several of the questions relate to a couple of key issues, so we take this opportunity to address all of them here. One issue that can certainly cause confusion is the use of both “dipolar” and “monopolar” to refer to the same quantity. The quantity in question is $\mathbf{L}_{1,\alpha}$, which is rotationally symmetric in spin space (as required in the zero spin-orbit coupling limit and when approaching the ordered phases from the paramagnetic side) and has an index α that is a spatial index:

$$\mathbf{L}_{1,\alpha} = \int r_\alpha \mathbf{s}(\mathbf{r}) d^3 r. \quad (\text{R2})$$

The integrand is the *dipolar* component of the local magnetization density $\mathbf{s}(\mathbf{r})$. In this paper, we wish to generalize altermagnetism from the collinear case (at zero spin-orbit coupling), where there is a globally conserved spin projection, to a non-coplanar spin texture where $\mathbf{s}(\mathbf{r})$ can be expected to rotate smoothly, spanning a spherical surface. Let us expand this in components

$$\mathbf{s}(\mathbf{r}) = \left(c^{\mu(0)} + c_\alpha^{\mu(1)} r_\alpha + c_{\alpha\beta}^{\mu(2)} r_\alpha r_\beta + \dots \right) \hat{\mathbf{n}}_\mu \quad (\text{R3})$$

where $\hat{\mathbf{n}}_\mu$ are components in spin space. To leading order, the integral picks out contributions of the form

$$\mathbf{s}(\mathbf{r}) = \sum_{\mu,\alpha} c_\alpha^{\mu(1)} r_\alpha \hat{\mathbf{n}}_\mu \quad (\text{R4})$$

which preserves the symmetry in spin space. To make the interpretation more transparent, we rotate in spin space so that the spin and spatial components are locked, i.e., $\alpha = \mu$. There are three independent components $\mathbf{c}^{(1)}$, which may be unequal, but the assumption of a non-coplanar texture implies that all three components should be equal.

In the case of a collinear magnetic structure, the spin texture is forced to lie along a single axis. In this case, the same order parameter will have a single p-wave component. Thus, the referee is correct that this order parameter is potentially applicable to collinear magnets at zero spin-orbit coupling and without inversion symmetry.

I also have a question on the number of multipoles. In p. 3, the authors stated that there are three monopolar (or dipolar?) and three quadrupolar altermagnetic secondary order parameters. However, the equation for the spatial dipole $L_{1,\alpha}$ in p. 7 allows three components ($\alpha = x, y, z$). Thus, if $L_{1,\alpha}$ amounts to the monopolar altermagnetic secondary order parameter, the number of $L_{1,\alpha}$ does not match the number of monopolar altermagnetic secondary order parameters stated in p. 3. The number of quadrupolar altermagnetic secondary order parameters is mysterious to me. In p. 3, the authors stated that there are three quadrupolar altermagnetic secondary order parameters. However, Fig. 2 shows only one quadrupolar distribution of $\langle S_z \rangle$ in the momentum space. If the quadrupolar altermagnetic secondary order parameters amount to the quadrupole distribution of $\langle S_z \rangle$ in the momentum space, I don't understand why the number of the quadrupolar altermagnetic secondary order parameters differs from the number of the quadrupolar distribution of $\langle S_z \rangle$ in the momentum space.

We thank the referee for pointing out a small inconsistency in the text. While the order parameters are presented correctly, the correct statement is that there is a single dipolar and a single quadrupolar order parameter, and both have three components.

The reason for showing only $\langle S_z \rangle$ is that we follow the definition of the secondary order parameter in the $k_z = 0$ plane.

$$\mathbf{O}_\alpha^1 = \int d^3k f_\alpha \mathbf{s}(\mathbf{k}) \quad \text{with} \quad f_\alpha = (k_x, k_y, k_z),$$

$$\mathbf{O}_\alpha^2 = \int d^3k f_\alpha \mathbf{s}(\mathbf{k}) \quad \text{where} \quad f_\alpha = (k_x k_y, k_x k_z, k_y k_z),$$

For \mathbf{O}_α^1 , only $\langle S_x \rangle$ and $\langle S_y \rangle$ can couple to $f_1 = k_x$ and $f_2 = k_y$, also for \mathbf{O}_α^2 , only $\langle S_z \rangle$ couple to $f_1 = k_x k_y$.

The multipole order parameters $L_{1,\alpha}$ and $L_{2,\alpha}$ should couple to the primary order parameter Φ_α . What are their coupling structures?

All order parameters have the same T symmetry in the zero spin-orbit coupling limit, so the coupling between them is the simple inner product of their three components for all order parameters.

B. Questions about the Spatial Dipole and Hedgehog Spin Texture

In p. 7, it is stated that a hedgehog spin texture is expected on the basis of the spatial dipole $L_{1,\alpha}$. I understand that the dipole $L_{1,\alpha}$ may imply the relation $\mathbf{s}(\mathbf{k}) = -\mathbf{s}(-\mathbf{k})$. However, I don't understand how this relation leads to the hedgehog spin texture. Although the hedgehog spin texture satisfies $\mathbf{s}(\mathbf{k}) = -\mathbf{s}(-\mathbf{k})$, the relation does not necessarily imply the hedgehog spin texture. For instance, the p-wave spin splitting [8] also satisfies the relation but has a qualitatively different spin texture from the hedgehog spin texture. So, it is unclear whether the symmetry analysis guarantees the hedgehog spin texture. If so, I wonder what the main difference is in terms of the symmetry analysis that distinguishes p-wave altermagnets from Mn_3IrSi with the hedgehog spin texture. A related question is how broadly the results of this manuscript apply beyond Mn_3IrSi . Does the presence of a non-collinear chiral altermagnet always imply the existence of a hedgehog spin texture? If not, what key conditions determine whether a system falls into this classification? Providing a more generalized argument will strengthen the impact of the work.

This is a good question that highlights areas where we can improve the presentation in the text. The referee is correct that a p-wave spin texture leads to non-vanishing components of the dipolar order parameter. However, we should expect such a pure texture only when the magnetic structure is coplanar. In the problem we are considering, the magnetic structure is chiral and non-coplanar, so there is no globally distinguished axis. In the case considered here, we expect a monopolar spin texture, reflecting the fact that the spin is only locally specified in momentum space and is tied to the non-coplanarity of the underlying structure.

In answer to the second question, we expect the physics reported here to be quite general, with monopolar textures allowed for chiral non-collinear structures in the zero spin-orbit coupling limit. The illustration here is a single instance of this more general case. A complete classification or

materials search is beyond the scope of this study.

C. Questions about Chirality

The authors emphasize both noncollinearity and chirality. From the manuscript, I can see that the noncollinearity makes a qualitative difference from the collinearity. However, it is not clear to me what role the chirality plays. In p. 6, it is stated that “Non-collinear altermagnets stand out because they break this C2T and the chirality can therefore be manifest in momentum space. Specifically, this means that the real space part of the multipolar order parameter can be odd.” So, this statement seems to imply that the chirality is necessary to violate the constraint $\mathbf{s}(\mathbf{k}) = \mathbf{s}(-\mathbf{k})$. However, this constraint can be violated once the inversion symmetry is broken. Although the inversion symmetry breaking is a necessary condition for the chirality, it is not a sufficient condition since the chirality requires the breaking of additional symmetries. If the inversion symmetry breaking alone is sufficient to violate the constraint $\mathbf{s}(\mathbf{k}) = \mathbf{s}(-\mathbf{k})$, it is more appropriate to say that the noncentrosymmetry is needed instead of the chirality for the real space part of the multipolar order parameter to have odd components. By the way, it is stated in p. 15 that “Due to the stricter symmetry constraints in non-chiral altermagnets (AM), spatially odd multipole components are not allowed in either collinear or non-collinear magnetic structures [6].” So, this statement indicates that the chirality (instead of the noncentrosymmetry) is necessary for the odd components of the multipolar order parameters. Ref. [6] appears to be the source of this statement. However, I could not find any related discussion in Ref. [6]. Therefore, I request the authors to clarify the role of the chirality for the odd components.

The referee is correct that inversion symmetry breaking is sufficient to violate the constraint $\mathbf{s}(\mathbf{k}) = \mathbf{s}(-\mathbf{k})$. Let us consider the effect of mirror symmetry. In the absence of spin-orbit coupling, it is natural to think about mirrors acting on real space and not on spin space. A mirror plane reverses the sign of the coordinate perpendicular to the mirror, so that component in $r_\alpha \mathbf{s}(\mathbf{r})$ is forced to zero in the presence of that mirror. This, therefore, is incompatible with the hedgehog spin texture we observe on constant energy slices in the band structure.

D. Questions about the SOC dependence

A few lines above Sec. VI, it is stated that “large spin Hall . . . and Edelstein . . . effects co-exist and have been calculated, showing very weak SOC dependence.” However, I don’t find any related discussion on the weak SOC dependence in Sec. IV, *Spin Hall and Edelstein Effects*. The only place where the SOC dependence appears is the caption of Fig. 3, which states that the light dashed lines in Fig. 3(a) and (b) are obtained for the model Hamiltonian without SOC. To make the claim of the weak SOC dependence convincing, I recommend the spin Hall effect and the Edelstein effect in Mn_3IrSi to be calculated through the first-principles calculations and explicitly compared for the cases with and without SOC.

We thank the referee for the comment and suggestion regarding the transport calculation without SOC. We also performed the Wannier projection based on the VASP software with infinite speed of light. The band structures from DFT and the Wannier-based tight-binding model are presented in Fig. R5, which shows good agreement around the Fermi level. Based on this Wannier-function model, the transport results were calculated. As shown in Fig. R6, the independent components discussed in the main text ($\sigma_{xy}^z, \sigma_{xz}^y, \chi^s$) are compared using dashed lines for calculations with SOC and solid lines for those without SOC. The magnitude and trends as a function of Fermi energy are consistent with and without SOC, as shown in Fig. R6.

FIG. R5: Band structure without spin-orbit coupling from DFT calculations (red dots) and projected Wannier functions (black solid lines).

FIG. R6: Spin Hall (upper panel) and Edelstein (lower panel) effect conductivity tensor elements versus energy around E_f . Red and blue denote σ_{xy}^z and σ_{xz}^x in the SHE, respectively. Dashed and solid lines correspond to calculations with full SOC and without SOC, respectively.

E. Questions about the relationship between odd-parity multipoles and odd-parity spin splitting (hedgehog spin texture)

In collinear d-wave altermagnets, real-space magnetic octupoles ($r^m r^n S^q$) are directly related to the k -space spin-even parity splitting ($k^m k^n S^q$), which is natural considering that they transform the same way under the time-reversal and also under the space inversion. However, the connection between odd-parity multipoles ($L_{1,\alpha}$ for example) and odd-parity spin splitting (hedgehog spin texture) is not obvious since these two quantities transform differently under the time-reversal (T) and under the inversion combined time-reversal (PT). For example, in Mn_3IrSi , flipping all spin directions (applying T) inverts $L_{1,\alpha}$, but does not invert the hedgehog spin texture. Are these two quantities directly related, or do they exhibit a more subtle dependence? A detailed discussion clarifying this point would be helpful.

We thank the referee for the question. The real-space odd-parity multipole changes sign under both time reversal and parity. These properties must be inherited by the order parameter in momentum space, where parity and time reversal separately take \mathbf{k} to $-\mathbf{k}$, and the sign reversal is

consistent with the hedgehog texture. In passing from real to momentum space, it is important to take $r_\alpha \rightarrow ik_\alpha$ to maintain the time-reversal properties, and this is likely the subtlety the referee refers to.

I have additional comments and questions, which are relatively minor compared to those given above.

It is stated that “The Landau theory further exemplifies an important aspect of collinear altermagnets namely that the order parameter breaks down a paramagnetic spin symmetry group to a collinear spin group.” In this statement, does the “order parameter” refer to the primary order parameter Ψ or the secondary order parameter O_Γ ?

It turns out that the relevant order parameters for altermagnetism—the (primary) Néel order parameter and the altermagnetic (secondary or pseudo-primary) order parameter—have the same symmetry. Therefore, both order parameters break the symmetry in the same way.

It is stated that “Non-collinear altermagnets stand out because they break this C_2T and ...” I understand this statement for noncoplanar magnets. However, for coplanar magnets, isn’t C_2T preserved?

The referee is entirely correct that in coplanar altermagnets the C_2T symmetry is present. We have updated the manuscript to correct this point. The intended meaning was to refer to non-coplanar magnets or, in the context of this paper, non-collinear, chiral systems.

In p. 7, the definition of \mathbf{M} is given:

$$\mathbf{M} = \sum_{i \in \text{prim}} \sum_{a=1}^2 \mathbf{S}_i^a.$$

I don’t understand index i in this definition. What does it refer to?

In this expression, we intend the sum to run over all magnetic sites. However, for the purposes of the symmetry analysis, it is helpful to think of this as the sum over all primitive unit cells and the sum over the basis within each unit cell. That is the meaning of the equation, and we have updated the manuscript to clarify this point.

In p. 7, it is stated that “ $\forall \alpha, \Phi_\alpha = 0$ ” for the IR E, where

$$\Phi_\alpha = \sum_{a=1}^{12} \phi_{E,a}^\alpha \mathbf{S}_a.$$

I don’t understand this statement. What is meant by “ $\forall \alpha, \Phi_\alpha = 0$ ” when there is only one IR E?

The E irrep has two components, and the α index runs over these, i.e., $\alpha = 1, 2$. The point here is that the magnetic structure does not belong to the E irrep, being orthogonal to both E components. We have rephrased this point to make it clearer.

In p. 7, the expression for the Landau theory reads,

$$F = c_2 \Phi_\alpha \cdot \Phi_\alpha + c_4 (\Phi_\alpha \cdot \Phi_\alpha)^2.$$

In this expression, is the summation over α assumed? If so, should the coefficients c_2 and c_4 be independent of α ?

Indeed, a sum over α is implied here, and the coefficients c_2 and c_4 are α -independent. We have added a remark to make this clear in the revised version.

In p. 7, the formula for the spatial dipole $L_{1,\alpha}$ reads

$$L_{1,\alpha} = \int d^3r r_\alpha \mathbf{m}(\mathbf{r}).$$

How does $\mathbf{m}(\mathbf{r})$ compare with $\mathbf{o}(\mathbf{r})$ in the formula for O_Γ in p. 5? Are they the same? If so, I recommend the authors use the same variables to avoid possible confusion.

We thank the referee for spotting this redundancy in the text. They are the same, and we now use a single notation.

In p. 8, it is stated that “The spin-symmetric basis in the T IR now breaks into three copies of the A IR.” Please clarify this statement. Considering that there are three T IRs in the absence of SOC, does it mean that each T IR in the absence of SOC breaks into three copies of A IR in the presence of SOC?

We thank the referee for the question. There are three T irreps for general magnetic structures in the zero SOC limit. However, the observed magnetic structure spans only one of these. This T irrep breaks into three A irreps in the presence of SOC.

Equation (2) in p. 11 contains the hopping coefficient $t_{i,j}$. How is it related to $(t_i, i = a, b)$ given five lines below Eq. (2)?

We thank the referee for pointing this out. To label the nearest and next-nearest hoppings, we use t_a and t_b instead, for simplicity. This clarification has been added to the text.

In the 2nd paragraph in Sec. IV, it is stated that “These therefore directly couple to the squared order parameter in the presence of SOC as the order parameter also transforms like A.” Does the order parameter refer to the primary order parameter or the secondary order parameter? Similarly, in the 3rd paragraph in Sec. IV, it is stated that “Instead, the Edelstein effect is cubic in the order parameter as . . .” Does the order parameter refer to the primary order parameter or the secondary order parameter?

The answer to the referee’s question is that *both* order parameters couple identically, as they have the same symmetry. The referee is essentially pointing out that we are using “secondary order parameter” in a non-standard way. Usually, “secondary” refers to an order parameter in a different irrep than the primary order parameter. Here, our secondary order parameter has the same symmetry. In the older literature, this is called “pseudo-primary.” We use “secondary” in a slightly different sense—the primary order parameter is the dipolar magnetic structure visible to, e.g., neutron diffraction, while the secondary order parameter is less directly observable. We are keen to preserve this terminology, but we have now highlighted that they have the same symmetry and therefore couple identically.

-
- [1] Y. Ōnuki, Y. Kaneko, D. Aoki, A. Nakamura, T. D. Matsuda, M. Nakashima, Y. Haga, and T. Takeuchi, *J. Phys. Soc. Jpn.* **91**, 065002 (2022).
- [2] A. E. Hall, P. Manuel, D. D. Khalyavin, F. Orlandi, D. A. Mayoh, L.-J. Chang, Y.-S. Chen, D. G. C. Jonas, M. R. Lees, and G. Balakrishnan, *Phys. Rev. Mater.* **7**, 114402 (2023).
- [3] N. Furukawa, *J. Phys. Soc. Jpn.* **64**, 2754–2757 (1995).

- [4] T. Eriksson, R. Lizárraga, S. Felton, L. Bergqvist, Y. Andersson, P. Nordblad, and O. Eriksson, Phys. Rev. B **69**, 054422 (2004).

[1] ϕ_{T^1} , ϕ_{T^2} , ϕ_{T^3} at the bottom of p. 7 and at the beginning of p. 8 seem to assume a specific labeling of the 12 magnetic atoms ($a=1, 2, 3, \dots, 12$) in a unit cell. I suggest that the authors provide these labels in Fig. 1(a) to help readers understand the physical meaning of the IR T.

Answer: Thanks for the suggestion. The corresponding labels have been added.

[2] In Eq. (4), the symbol ϕ is used in ϕ_{A1} , ϕ_{A2} , ϕ_{A3} to denote the IR A. However, in the earlier part of the manuscript, the symbol Φ (uppercase) is used to denote IRs and the symbol ϕ (lowercase) to denote the basis of IRs. For notational consistency, I recommend that the authors use Φ (uppercase) to denote the IR A in Eq. (4).

Answer: Thank you for pointing this out and it has been revised.

[3] 1st line in p.12: It was stated that α denotes the sublattice. However, the third term in the Hamiltonian H [Eq. (5)] seems to imply that α is the spin index. I request the authors to clarify this point.

Answer: Thanks. The Hamiltonian [Eq. (5)] represents a general form that includes hoppings, spin-orbit coupling, and local magnetic moments. To investigate the properties of Mn₃IrSi, we adopt a specific basis associated only with the sublattice and spin degree of freedom. For the consistency of notation, we revised the label of basis to $|i, \sigma\rangle$, where i and σ denote the sublattice and spin degree of freedom, respectively.

[4] 4th line in the second paragraph at p. 13: Does ϵ_0 amount to $\epsilon(k_0)$? If so, I suggest that the authors replace ϵ_0 with $\epsilon(k_0)$.

Answer: Yes, and it has been revised.

[5] 2nd line from the bottom of p.13: It was stated that “For the crystal symmetry of Mn₃IrSi, the spin Hall tensor σ^i_{jk} has two independent time-reversal even, non-vanishing components: σ^z_{xy} and σ^y_{xz} , each ...” I was confused by this statement. If I ignore the magnetic moment arrangement and focus on the crystal symmetry, the 90-degree rotation around the x-axis followed by a proper fractional translation is a nonsymmorphic symmetry of the crystal structure and forces the two components to have the same magnitude with opposite signs. So I guess what the authors intended is the magnetic symmetry instead of the crystal symmetry.

Answer: Thanks for the comments. The magnetic space group is in fact isomorphic to the space group, and the magnetic ordering doesn't break the symmetries without magnetism but only the time-reversal symmetry. The two-fold rotational nonsymmorphic symmetry does flip the sign, but this happens only for the component σ^i_{jk} where two or three of the indices are identical, such as σ^x_{yy} (under C_{2y} : $\sigma^x_{yy} \rightarrow -\sigma^x_{yy}$) or σ^x_{xx} (under C_{2y} : $\sigma^x_{xx} \rightarrow -\sigma^x_{xx}$). Whenever symmetry requires an opposite sign, the corresponding component is constrained to vanish. For σ^z_{xy} , any two-fold rotational symmetry flips two of the indices and leave it invariant.

[6] p.14: It was stated in the paragraph right below Eq. (6) that only the second expression in Eq. (6) is relevant for the intrinsic spin Hall effect and only the first expression is relevant for the Edelstein effect. On the other hand, the description right above Eq. (6) implies that

both expressions contribute to both the spin Hall effect and the Edelstein effect. I am afraid that this inconsistency may cause confusion to readers. I suggest that the authors revise this part of the manuscript to avoid possible confusion.

Answer: Thanks for raising this point. We have added a clarification in this part. In principles, both intrinsic and extrinsic part contribute to the observables, with the difference being whether they involve the material-dependent scattering time. As we stated in the main text, two response tensors (intrinsic & extrinsic) share the same form. For the spin Hall effect, we have included the extrinsic results in the supplementary materials. For Edelstein effect, it is common to compare the experimental results with the extrinsic since it has the relative larger magnitude [Nano Lett. 19, 5959-5966 (2019), Nat. Comm. 15, 7663 (2024)].